# Knee moments of anterior cruciate ligament reconstructed and control participants during normal and inclined walking

Raghav K Varma, Lynsey D Duffell, Dinesh Nathwani, Alison H McGregor

MSK Lab, Imperial College London, London, UK

**Correspondence to**
Professor Alison McGregor;
a.mcgregor@imperial.ac.uk

## ABSTRACT

**Objectives:** Prior injury to the knee, particularly anterior cruciate ligament (ACL) injury, is known to predispose one to premature osteoarthritis (OA). The study sought to explore if there was a biomechanical rationale for this process by investigating changes in external knee moments between people with a history of ACL injury and uninjured participants during walking: (1) on different surface inclines and (2) at different speeds. In addition we assessed functional differences between the groups.

**Participants:** 12 participants who had undergone ACL reconstruction (ACLR) and 12 volunteers with no history of knee trauma or injury were recruited into this study. Peak knee flexion and adduction moments were assessed during flat (normal and slow speed), uphill and downhill walking using an inclined walkway with an embedded Kistler Force plate, and a ten-camera Vicon motion capture system. Knee injury and Osteoarthritis Outcome Score (KOOS) was used to assess function. Multivariate analysis of variance (MANOVA) was used to examine statistical differences in gait and KOOS outcomes.

**Results:** No significant difference was observed in the peak knee adduction moment between ACLR and control participants, however, in further analysis, MANOVA revealed that ACLR participants with an additional meniscal tear or collateral ligament damage (7 participants) had a significantly higher adduction moment (0.33±0.12 Nm/kg m) when compared with those with isolated ACLR (5 participants, 0.1±0.057 Nm/kg m) during gait at their normal speed (p<0.05). A similar (non-significant) trend was seen during slow, uphill and downhill gait.

**Conclusions:** Participants with an isolated ACLR had a reduced adductor moment rather an increased moment, thus questioning prior theories on OA development. In contrast, those participants who had sustained associated trauma to other key knee structures were observed to have an increased adduction moment. Additional injury concurrent with an ACL rupture may lead to a higher predisposition to osteoarthritis than isolated ACL deficiency alone.

## Strengths and limitations of this study

- To the best of our knowledge, this is the first report looking at external moments during inclined and declined walking for anterior cruciate ligament (ACL) reconstruction participants.
- In addition to looking into the external moments of the affected and unaffected knee, this study also looked at the effect of gait speed on external moments and differences in Knee injury and Osteoarthritis Outcome Score (KOOS) for each group.
- This study provides a potential explanation for the disparity seen in previous studies looking into peak knee adduction moment in ACL reconstruction (ACLR) and matched control participants.
- This study suggests that injuries to other key knee structures may play a bigger part in inducing osteoarthritis than ACL injury alone.
- One limitation to this study is the small sample size, in particular after dividing our ACLR group into ACLR+ and ACLR− groups.

## INTRODUCTION

Anterior cruciate ligament (ACL) injuries are common, exceeding 100 000 annual cases in the USA.[1] The majority are sports-related injuries, and lead to knee instability as a result of increased anterior tibial translation and anterolateral rotation.[2] ACL reconstruction (ACLR) is the primary treatment for an ACL rupture and permits return to a range of high-level activities including sport. It is accepted that people with ACL injuries, including those who undergo surgical reconstruction, are prone to further knee degeneration[3] and early osteoarthritis (OA).[4][5] Lohmander et al[3] reviewed 127 publications and determined that the overall mean incidence of developing OA after an ACL injury with/without reconstruction is over 50%[3] with the majority noting radiographic signs of OA 10 years after injury. Gait biomechanics

are considered to play a vital part in knee joint degeneration,[5][6] with altered kinematics and kinetics changing the distribution of mechanical load on the knee.[7] This in turn is postulated to lead to cartilage wear[5–7] and eventually knee osteoarthritis.

There is consensus among researchers that ACL deficient patients employ different gait strategies.[8] In vivo studies have found reduced knee flexion,[5] increased internal tibial rotation[5][9] and increased knee adduction moment[10] during level walking to be the three main changes in external knee moments following an ACL rupture. Furthermore, research has indicated that ACL reconstruction does not restore normal knee mechanics.[11] Berchuck *et al*[12] in 1990 noted reduced knee flexion during normal gait, indicating a coping strategy termed quadriceps avoidance gait. Anterior displacement of tibia through the contraction of quadriceps is balanced by the ACL when the knee is at an angle of 0–45°.[12] People with ACL rupture and/or ACLR are found to have quadriceps activation deficits,[13] which may be due to a central regulatory mechanism to avoid further joint damage by these muscle groups. Gait adaptations in the sagittal plane can lead to knee joint instability and ligament laxity.[14] This may result in osteoarthritis initiation and progression.[14]

High moments in frontal and transverse planes of the knee have been linked to OA.[5] ACLR has been shown to restore rotational stability,[9][15] however high-knee adduction moments (KAM) after reconstruction have been observed[16] but such changes are not universally agreed.[17][18] This is of particular importance since a 1% increase in adduction moment at the knee is thought to increase the risk of knee OA by 6.5 times.[19] The discrepancies in previous studies may be due to different walking speeds, and higher KAM may only be evident during more challenging tasks. Hence in this study we aimed to gain a better understanding of peak knee moments in the frontal and sagittal planes during gait at different speeds and inclines. Our primary aim is to compare peak knee moments in the sagittal and frontal planes of ACLR participants with healthy controls on sloped surfaces, with a view to explore the biomechanical basis for the observation that ACL injury predisposes one to OA. Our secondary aim was to investigate the effect of speed on peak moments. Finally, we compared functional outcome scores between groups using the Knee injury and Osteoarthritis Outcome Score (KOOS).

## METHODS

This cross-sectional study explored peak knee moments between ACLR and healthy control participants during inclined walking. The study was approved by the Imperial College Research Ethics Committee. We used the Strengthening the Reporting of Observational studies in Epidemiology statement as a checklist for our observational study.[20]

A total number of 24 participants participated in this study and written informed consent was obtained; details are provided in table 1. The ACLR inclusion criteria were: aged between 18 and 60 years; body mass index (BMI) <30 kg/m²; a complete, unilateral ACL rupture followed by a single bundle hamstring autograft reconstruction that was performed at least 1 year ago with no history of knee trauma or injury to their contralateral leg. Participants who were unable to walk comfortably on a 10° incline walkway were also excluded. The control group did not have any muscular or neurological lower limb pathology and were matched to the ACLR participants with respect to gender, activity, height, weight and their dominant leg (leg preference for kicking). All participants completed the KOOS.[21] We measured the participants' activity levels using Tegner activity scale.[22]

A three-dimensional motion analysis system (Vicon MX T-20 System, Vicon, Oxford, UK) was used to collect kinematic data for normal, slow, upslope and downslope gait. This software used ten-motion capture cameras to pick up 35 reflective markers at a sampling rate of 100 Hz. The reflective markers were placed bilaterally on the head of the second metatarsal, head of the fifth metatarsal, head of talus, calcaneal tuberosity, medial and lateral malleolus, medial and lateral femoral epicondyle, anterior superior iliac spine, posterior superior iliac spine, acromion and one single marker on the manubrium. Marker clusters (3 reflective markers on each) were affixed bilaterally to the calf and thigh. Kinetic data (ground reaction force) were collected using portable force plates (Kistler Instruments AG, Winterthur, Switzerland) at a sampling rate of 1000 Hz and were synchronised with the camera data.

The participants were asked to walk barefoot. A 5 min self-directed warm up allowed the participants to familiarise themselves with each task.

A 7 m long walkway was used, 2.5 metres of which could be raised to form a ramp, at an incline of 10°. It

**Table 1** Participant characteristics, activity level and time since surgery

|  | ACLR (SD) | Control (SD) | Unpaired t test |
|---|---|---|---|
| Age (year) | 30.5 (8.68) | 24.8 (8.81) | p=0.125 |
| Height (m) | 1.76 (0.13) | 1.73 (0.11) | p=0.547 |
| Weight (kg) | 75 (11.13) | 71.6 (11.2) | p=0.464 |
| Tegner activity scale | 6.25 (1.82) | 6.08 (1.93) | p=0.826 |
| Time since surgery (year) | 4.5 (3.5) | NA |  |

Twelve participants in ACLR group and 12 in control group.
ACLR, anterior cruciate ligament reconstruction; NA, not applicable.

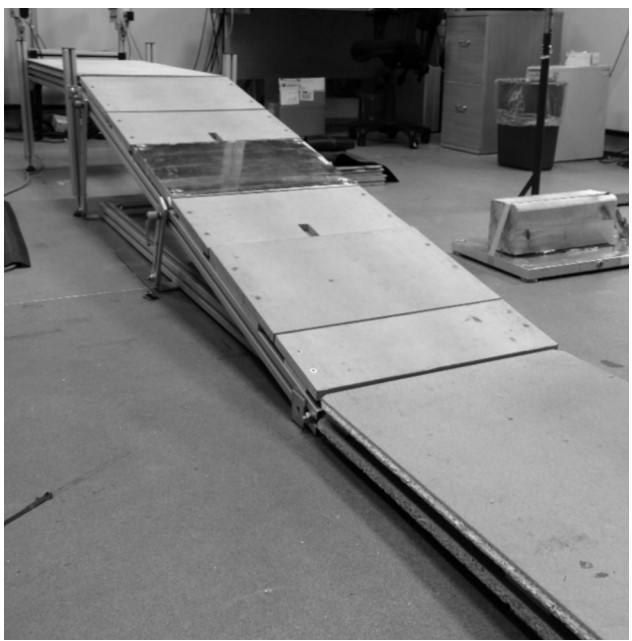

**Figure 1** Steel-framed ramp covered in plywood, set an incline of 10°.

was constructed from a steel frame and covered with plywood, with one portable force plate embedded in the centre (figure 1). Participants were asked to walk at a self-selected pace uphill and downhill. The ramp was then removed to create a level walkway. All the participants were asked to walk at a self-selected pace and at a pace which they considered to be slow. Each task was repeated until both feet made complete contact with the middle of the force plate at least three times.

All data were time normalised to one gait cycle. The data were analysed from the stance phase of the gait cycle, when the ground reaction force reached more than 40N (heel strike) to when it dropped to less than 40N (toe off). A fourth order Butterworth Filter at a cut off (12 Hz) was used to reduce noise. Joint angles and

moments were calculated from the position of the reflective markers and the ground reaction force data using a custom model written in bodybuilder software.[23–25] Peak moments in the sagittal and frontal planes of the knee were extracted using MATLAB (R2013b) software.

Unpaired Student t tests were used to determine any significant differences in demographics between ACLR and control group. Multivariate analysis of variance test was used to calculate significant differences in all other parameters. Tukey honest significant difference approach was used to establish significance, with the α value set at 0.05. All statistics were carried out using SPSS V.22.

## RESULTS

Twelve ACLR participants (9 men and 3 women) and 12 controls (9 men and 3 women) were recruited for this study. There were no significant differences between the groups in age, height, weight and Tegner activity scale (table 1). The mean time since reconstruction surgery was 4 years and 6 months.

We further divided our ACLR group into two: participants that had additional cartilage, meniscus or ligament damage in their ACLR leg (ACLR+ group; 7 participants) and participants with isolated ACL injuries (ACLR− group; 5 participants). The additional knee injuries to the ACL rupture are meniscal tear (3 participants), cartilage damage (3 participants) and torn medial collateral ligament (MCL; 1 participant).

No statistically significant differences were found in peak knee adduction moment between ACLR and control participants during uphill and downhill gait, and during gait at normal and slow walking speeds on a flat surface (figure 2). Further analysis revealed that ACLR participants with meniscal tear, cartilage damage or MCL damage (ACLR+) had significantly higher knee adduction moment (0.33±0.12 Nm/kg m) during gait on a flat surface at a normal walking speed compared

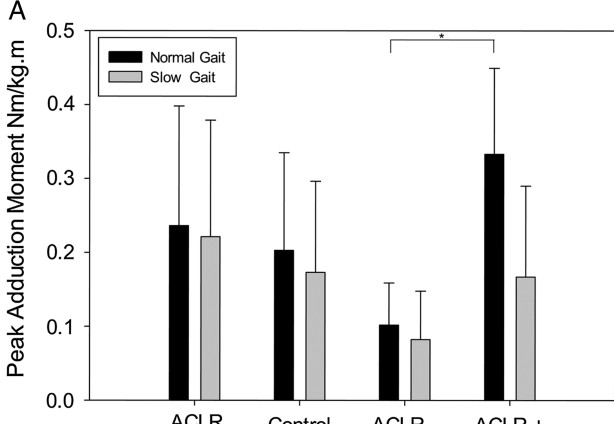

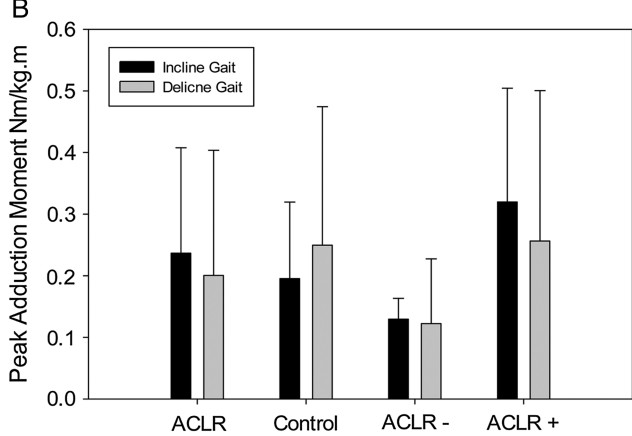

**Figure 2** Peak adduction moments in (A) level walking and (B) inclined walking for ACLR, ACLR+, ACLR− and control group. Asterisk indicates significance (p=0.042).

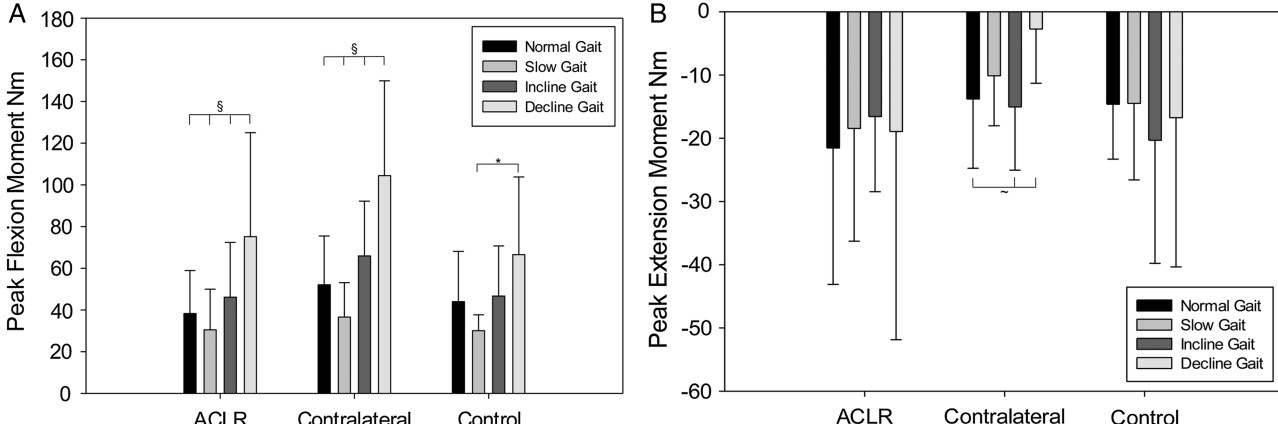

**Figure 3** (A) Peak flexion moment in all activities for ACLR, contralateral and control group. §Represents decline gait to be significantly higher than all other activities (p<0.01). Asterisk indicates significance, p<0.05 (B) Peak extension moment in all activities for ACLR, contralateral and control group. ~Represents decline gait to be significantly lower than normal and incline gait, p<0.05.

with those participants with an isolated ACL injury (ACLR−, 0.1±0.057 Nm/kg m), p=0.042 (figure 2).

This was not the case for data collected in the sagittal plane. There was a tendency for the contralateral (unaffected) knee of ACLR participants to show a higher mean knee flexion moment in all activities compared with ACLR affected knees and control knees (figure 3). The difference was not found to be statistically significant.

Table 2 shows that there were no significant differences in gait speed in normal or slow walking between any groups. The control group had significantly higher scores in all of the KOOS domains apart from activities of daily life compared with the ACLR group (table 3). No significant KOOS differences were seen between ACLR+ and ACLR− groups.

## DISCUSSION

In the frontal plane we found no statistically significant differences between our ACLR participants and controls. However ACLR participants who had sustained associated trauma to other key knee structures (meniscal, collateral ligament and chondral damage) were observed to have a higher adduction moment during gait on a flat surface at normal walking speed when compared with participants with isolated ACLR. In the sagittal plane there was a tendency for ACLR participants to

have higher peak knee flexion moment in their contralateral leg during all activities.

Previous studies that have investigated peak knee adduction moment in ACLR and matched control participants have provided mixed results.[16–18] We investigated similar and more challenging gait set-ups (by altering incline), as well as the effects of differences in walking speed, in order to explore the biomechanical basis for the observation that ACL injury predisposes one to OA. Our data showed no significant differences in KAM between our ACLR participants and control participants under all conditions. This indicates that providing more challenging gait set-ups such as inclined walking, where a higher range of motion in the sagittal plane is required, does not emphasise differences between ACLR and control participants. Based on our findings, the discrepancies in previous studies appear not to be related to the difficulty of the task or differences in walking speed.

ACL injury is often accompanied by other knee injuries.[26] Prevalence of associated meniscal damage and chondral lesions at the time of ACL injury can be as high as 65% and 23%, respectively.[27] Associated knee injuries are thought to increase the incidence of OA from 0–13% in isolated ACL injury to 21–48%.[28] This may be because key knee structures such as menisci prevent cartilage wear by distributing loads and functioning as shock absorbers.[14] Our results suggest that

**Table 2** Gait speed during normal and slow, level walking tasks

|  | ACLR | Control | ACLR+ | ACLR− | p Value |
|---|---|---|---|---|---|
| Gait normal speed | 1.17 (0.13) | 1.20 (0.11) | 1.18 (0.15) | 1.16 (0.11) | 0.940 |
| Gait slow speed | 0.76 (0.13) | 0.75 (0.11) | 0.78 (0.16) | 0.74 (0.09) | 0.885 |

Data are mean (SD).
ACLR, anterior cruciate ligament reconstruction; ACLR+, participants with other knee injuries in their ACLR leg; ACLR−, participants with isolated ACL injuries.

**Table 3** Knee injury and Osteoarthritis Outcome Score (KOOS) with SD for each domain recorded for each group

| KOOS outcome | ACLR (SD) (n=12) | Control (SD) (n=12) | ACLR+ (SD) (n=7) | ACLR− (SD) (n=5) | Significant values |
|---|---|---|---|---|---|
| Pain | 88.4 (9.32) | 99.1 (3.2) | 87.5 (8.83) | 89.4 (10.8) | Control vs ACLR: p=0.010 Control vs ACLR(+):p= 0.019 |
| Symptoms | 83.1 (11.4) | 98.2 (3.19) | 85.1 (12.7) | 80.7 (10.6) | Control vs all other groups: p<0.05 |
| Activities of daily life | 96.3 (5.63) | 100 (0) | 98 (3) | 94.4 (7.7) | No significant differences |
| Sport and recreation | 83.8 (16.9) | 99.6 (1.4) | 89.1 (7.4) | 77.4 (23) | Control vs ACLR: p=0.006 Control vs ACLR(−):p=0.003 |
| Knee-related QOL | 64.5 (23.2) | 100 (0) | 64.6 (23.3) | 70 (26.7) | Control vs all other groups: p<0.05 |

Data are mean (SD).
ACLR, anterior cruciate ligament reconstruction; ACLR+, participants with other knee injuries in their ACLR leg; ACLR−, participants with isolated ACL injuries.

people with ACLR injuries with associated knee injuries experience higher knee adduction moments than people with isolated ACLR injuries.

Of the three previous studies that have looked at peak KAM in ACLR and matched control participants, Butler et al[16] found the ACLR group to have a higher peak KAM compared with controls. This was also seen in our ACLR+ group. In the studies that followed, Webster and Feller[17] and Zabala et al[18] found the ACLR group to have a reduced peak knee adduction moment to controls. This was seen in our ACLR–group (figure 2). This disparity in studies may therefore be a consequence of different exclusion criteria for ACLR participants. Butler et al[16] did not exclude ACLR participants with other knee injuries, while the other two studies excluded participants with ligament damage[17 18] and also those with >25% of menisci loss.[18] We suggest that associated knee injuries are related to increased knee adduction moments in ACLR participants.

We found the difference between peak KAM in ACLR+ and ACLR− to be statistically significant only during normal gait. We expected the difference to be higher during inclined walking, as it is more challenging than level walking. Change in terrain, muscle weakness, gait deficit and balance deficit are primary risk factors for falling.[29 30] This suggests that ACLR participants need to adopt a conservative gait strategy while walking on a sloped surface to ensure safety. During challenging tasks such as downhill walking, healthy participants increased their metabolic activity and implemented a conservative gait strategy to reduce the risk of falling.[29] This principle may also be applied by ACLR participants to ensure safety.

In the sagittal plane, no statistically significant differences were observed between ACLR, ACLR+, ACLR− and control group in peak knee flexion and extension moments. Uphill and downhill walking require greater use of quadriceps muscle than on a level walkway.[30 31] Two years after an ACLR surgery, differences in quadriceps strength between limbs are no longer seen.[29] All of the ACLR participants had undergone reconstruction at least 1 year before taking part in this study, with an average of 4.5 years. This indicates that all participants may have had sufficient time to restore their quadriceps strength. Although sagittal instability is thought to increase joint loads and lead to joint failure,[14 32] it may not play a significant role in OA induction and progression after reconstruction.

An unexpected finding was the tendency for the contralateral knees of ACLR participants to have higher peak knee flexion and lower peak knee extension moment compared with the ACLR and control knees in all activities. This may be an adaptation to reduce loading on their ACLR knee. Patients with advanced knee OA also display this adaptation to reduce loading on their injured leg.[33] Although this may present as a mechanism to slow the progression of OA, there are harmful implications associated with the contralateral leg. Weight-bearing asymmetry may induce OA in the contralateral leg[34]; 37% (24/65 female patients) showed signs of radiographic OA in their contralateral leg, 12 years after ACLR.[35] This suggests that in this population unilateral injury changes joint function bilaterally.

In addition to our primary investigations we also investigated differences in KOOS and walking speed. There was no significant difference in KOOS or gait speed between ACLR+ and ACLR−. This indicates that high peak knee adduction moment in the injured leg does not affect our participants' pain outcome, symptoms, activities of daily life, sport and recreation, knee-related quality of life and gait speed. Therefore, patient-reported outcome measures and gait speed might not provide the clinician with any information about different gait adaptations.

Mundermann et al[36] found a 10.2% reduction in maximum knee adduction moment when people with less severe OA reduced their walking speed from 1.2 to 0.8 m/s. However, in the current study the difference in peak knee moments between normal (1.17 m/s) and slow gait speed (0.76 m/s) was not statistically significant. This may be due to our small sample size.

It is important to note that different knee injuries, rehabilitation protocols and time between injury and reconstruction are all thought to influence joint moments.[18] These are limitations that should be

considered when examining the results presented here. Additionally our participants were not recruited straight after their ACLR; some participants may have had further injury or pathological changes within the joint since the reconstruction. Another limitation was our small sample size, in particular after dividing our ACLR group into ACLR+ and ACLR− groups.

The ramp was set at an incline of 10° because the transition from a level to inclined walking strategy is thought to be around 5.5°[31] and after an incline of 10° no kinematic differences are seen in healthy participants.[31]

With regard to Vicon Motion capture system, different skin marker placement and skin motion artefacts are thought to increase error.[6] We tried to reduce the effects of different skin marker placement by having only one researcher place all markers on each individual. In addition we used a model that used clusters to reduce the effects of skin motion.[25]

## CONCLUSION

In conclusion, this study found no significant differences in peak moments in the frontal and sagittal planes during level and inclined walking for ACLR compared with control participants. However, we noted that individuals who have other knee injuries associated with their ACLR knee exhibit higher peak adduction moments during level walking at their normal speed. This suggests that injuries to other key knee structures may play a bigger part in inducing OA than ACL injury alone, although this requires further investigation with a larger sample size. Our data also suggest that the contralateral knee appears to be functioning in such a way to reduce high moments in the ACLR knees, which may be relevant in the risk of OA development in both knees. These findings warrant a longitudinal study comparing the knee adduction moment between isolated ACLR injury and ACLR with additional knee injuries and the prevalence of premature OA.

**Acknowledgements** The authors acknowledge support from Mr R Weinert-Aplin for the facilitation of the instrumented inclined walkway and conversion of the output data. He was also involved in designing the data collection tool.

**Contributors** RKV carried out data collection, statistical analysis, analysis of the data produced. He also drafted, revised the manuscript and is the guarantor. LDD attained ethical approval, determined the methodology, helped with data analysis, drafted and revised the paper. DN helped with data collection and drafted and revised the paper. AHM monitored the data collection for the whole trial, provided the data collection tools, drafted and revised the paper.

**Funding** The authors acknowledge the support from the Medical Engineering Solutions in Osteoarthritis Centre of Excellence funded by the Wellcome Trust and EPSRC.

**Competing interests** None.

**Ethics approval** Imperial College Research Ethics Committee.

**Provenance and peer review** Not commissioned; externally peer reviewed.

**Data sharing** No additional data are available.

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
