## [Reviewer comments · BMJ Open]

Some articles will have been accepted based in part or entirely on reviews undertaken for other BMJ Group journals. These will be reproduced where possible.

ARTICLE DETAILS

TITLE (PROVISIONAL)	Knee moments of anterior cruciate ligament reconstructed and control participants during normal and inclined walking.
AUTHORS	Varma, Raghav; Duffell, Lynsey; Nathwani, Dinesh; McGregor, Alison

VERSION 1 - REVIEW

REVIEWER	Britt Elin Oiestad Department of Orthopaedics, Oslo University Hospital, Norway and Norwegian research center for Active Rehabilitation
REVIEW RETURNED	24-Feb-2014

GENERAL COMMENTS	1. The research questions in the abstract and introduction are different. Please be more specific on primary aim (gait outcomes) and secondary aims (function).2. See specific comments for abstract3. The study design is appropriate, but the sample is too small, too heterogen, and is not matched on important variables.7. I do not see why paired t-tests should be used. The samples are independent because they are not matched.8. I miss important gait studies on ACL injured patients.10. The result section is not emphasizing the most important results: gait outcomes11. The conclusions are not justified by the results because the time of the examination was 4.5 years after ACLR, which may suggest that some of the patients already have OA, and the gait abnormalities may be a result of OA and not acute injuries to the knee12. See specific comments13. The STROBE statement is included, but it is not mentioned in the manuscript. In general: Conducting gait analysis studies are demanding, and I acknowledge the authors for doing such studies. However, I am worried that the data is not valid for answering the aims of the study. The main concern is the small sample size. Please include post hoc sample size calculations that may reveal if there are included enough study participants. The second concern is the time point in which the examinations were done. 4.5 years after ACLR may be too long to see influence of acute knee injuries only. Onset of osteoarthritis may already influence the gait characteristics of the study participants. Third, the study objective, methods, results and discussion do not follow a logical outline. Please restructure the manuscript such that the aim of the study is presented first throughout the chapters. My third concern is that the sample is too heterogen to answer the research question. With small sample sizes, the sample should be
---

homogeneous. A better approach would be to include only people with isolated ACL injuries if you would go for such a small sample. Otherwise it is hard to detect associations. Furthermore, the controls should be matched on age, gender, BMI, and activity level.

Abstract:

Background: the aim of the study is unclear stated in the background. Please specify according to investigating changes in gait, and not knee function.

Methods: It would be informative if the gait measures were specified. Type of statistical methods?

Results: In the sentence "A similar, but non-significant trend was seen in all the other activities": please specify the activities.

Conclusion: the first part of the conclusion is based on 5 subjects with isolated ACL injury. Furthermore, only 7 subjects had additional injury. On the basis of the results presented in the abstract, the conclusion seems too bold.

In the article summary: new information in bullet 2 that is not mentioned in the abstract is given. This should be implemented in the method section and the results.

Introduction

Page 5, line 3: is the avoidance pattern only to prevent further ACL damage, or is it also because of quadriceps activation failure?

Page 5 line20: do the authors mean 1% increase in adduction moment (not "adduction of moment")?

Page 5, line 25-30: the aims stated here is different from the aims in the abstract. Please be consistent with the aims throughout the manuscript.

Methods: how was the study participants included? And from where? Which time point after ACLR were the study participants examined?

Page 5, lines 47-57. Please re-write to include more sentences and fill in missing words. I prefer subject characteristics in the beginning of the result section, and a presentation of methods only, in the method section.

Page 6, line 15: the control participants seem somewhat older. Why weren't the control subjects' matched on sex, age and activity level?

Page 6, line 23-37: Adding up the markers described does not give a total of 35 as indicated in line 28. Please be specific if the number 35 means the capacity of the camera or the amount of markers that were included in the study.

Page 6, line 38: why were reflective markers put on acromion and manubrium?

Page 7, line 40-50: please include power calculations for rationale of the amount of subjects included in the study.

Page 7, line 41: Why do you use paired sample t-test when there is no matching of the cases and controls? Are the sample correlated in any way?

Results

Page 8, line 6: why were the controls older than the cases? Or put another way, why weren't the controls matched on age? Please give an explanation.

Page 8, line 6-11: Which variable were included in the multiple regression analyses to assess the influence of age? Did both the controls and cases be accounted for in this analysis?

Page 8, line 13: the Tegner Activity Scale should be included in the method section as an outcome measure.

Page 8, line 36-44: The groups are very small. Give a rationale for why the sample is divided into groups.

Page 8, line 34: I suggest answering the aim of the study starting from here, and present less important results that are not stated as aims of the study in the end of the result section.

Page 9, Table 2: it is not clear what the column of «Post hoc analyses» means. Please explain under the table.

Page 9, Table 2: It is unclear what the p-value reflect (for which groups?). If the p-values reflect ACLR vs. controls, the table would be more visible if the p-values were moved to fourth column. The same goes for table 3. Please also consider to make a table for the gait variables as table 2 – the most important results should be presented before the functional data which are not specified in the objective of the study.

Discussion

I would suggest that the first paragraph of the discussion present the aim of the study, and the results of the study.

Page 10, line 37: the reference Claes et al. should be updated to Oiestad et al 2009, which is the original review of the numbers presented here.

The statement on page 10, line 42-47 «Our results suggest that high knee adduction moments may be a relevant risk factor in OA incidence in ACLR subjects only when associated knee injuries are present.», should be modified or deleted. This study have no data on osteoarthritis, thus, no association between knee adduction moments and OA can be made.

Line 55: where did the authors present data to underline the statement of a difference between ACLR+ and controls?

Page 10, line 55 and page 11, line 2: please replace “to controls” with “compared to controls”

Page 12, Line 38-39: use KOOS consistently

The written English should be updated on some parts of the manuscript.

The STROBE statement is attached, but it is not mentioned in the manuscript.

REVIEWER	Zi-Sheng Ai Tongji university,China
REVIEW RETURNED	14-Mar-2014

GENERAL COMMENTS	1 Please explain why not consider age, gender as matching factors? In Page8 Line 25, after age differences appeared in the two groups of Table1, Please explain the rationality of using One-way ANOVA in Table2 & Table3? 2 Statistical analysis are carried out using the SigmaPlot 11.0 on windows, please explain why not used commonly SAS, SPSS or Stata software? 3 In this study, the sample size is too low, the conclusion is not reliable, please calculate power of a test? 4 Table2 in Page9 Line 3, and Table3 Page9 Line40, there were one-way analysis of variance among four sample means, which ACLR, itself including ACLR + and ACLR-. It was wrong about carrying out one-way ANOVA of comparing four sample means & multiple comparison in statistics, please correctly analyze the data. 5 Furthermore, patients with ACL injury were divided into two subgroups. In ACLR+ group, we find there are 3 cases with associated meniscal tear, 3 with cartilage damage and 1 with torn MCL. That means, the ACLR+ group is heterogeneous. We do not know which associated injury is more important. Therefore, I think the conclusion is not supported totally by the methodology. 6 Originality The study of relationship between anterior cruciate ligament (ACL) reconstruction and kinetic during gait can help to elucidate the causes for high incidence of osteoarthritis after reconstruction. This paper tell us the effect of additional injury following ACL rupture on the osteoarthritis. 7 Language Good. However, needs a more careful proofreading. There are some grammar mistakes and in some points the content is confusing. Page 13 line 46 "artefacts " need a revision. 8 Previous Research: Previous research was referenced appropriately and the references are accurate. 9 Ethical Issues: No fraud will be discussed. 10 Recommendation: Accept but needs major revisions. This reviewer will be happy to review the revised article.
--

VERSION 1 – AUTHOR RESPONSE

Reviewer 1 (Britt Elin Oiestad)

-The research questions in the abstract and introduction are different. Please be more specific on primary aim (gait outcomes) and secondary aims (function).

•We have made our aims clearer and more specific. Page 2 line 14 – 22, page 5 line 37 – 52

-I do not see why paired t-tests should be used. The samples are independent because they are not matched.

•We apologise for our error, we have changed our statistical tests to unpaired t-test. Our results now show that age is no longer statistically different between the two groups. Table 1

-I miss important gait studies on ACL injured patients.

•We have included papers that we did not come across when writing the manuscript. Including - Gokeler A, Benjaminse A, van Eck CF, et al. Return of normal gait as an outcome measurement in acl reconstructed patients. A systematic review. Int J Sports Phys Ther. 2013 Aug;8(4):441-51. REF – 11

-The result section is not emphasizing the most important results: gait outcomes

•Agreed, we have moved gait outcomes to the beginning of the result section and hope this is clearer. Page 8 line 52

-The conclusions are not justified by the results because the time of the examination was 4.5 years after ACLR, which may suggest that some of the patients already have OA, and the gait abnormalities may be a result of OA and not acute injuries to the knee

•Ideally we would have recruited the participants just after their ACLR, however this was not achievable with our study design. We have added this limitation to the discussion. Page 14 line 16 – 21

-The STROBE statement is included, but it is not mentioned in the manuscript.

•The STROBE statement has been added to methods. Page 6 line 8 – 10

-Please include post hoc sample size calculations that may reveal if there are included enough study participants.

•Using Roy's Largest Root we found the power of the multivariate tests looking at gait outcome to be 0.838. However Butler et al. found that a meaningful change of 10% needed 15 subjects to examine differences in peak knee-abduction moment. (Butler RJ, Minick KI, Ferber R et al. Gait mechanics after ACL reconstruction: implications for the early onset of knee osteoarthritis. Br J Sports Med 2009;43:366-70.)

-Third, the study objective, methods, results and discussion do not follow a logical outline. Please restructure the manuscript such that the aim of the study is presented first throughout the chapters

•We hope the structure is now clearer

-A better approach would be to include only people with isolated ACL injuries if you would go for such a small sample. Otherwise it is hard to detect associations. Furthermore, the controls should be matched on age, gender, BMI, and activity level.

•We believe the ACLR+ group provide important results that, while only exploratory, warrant publication. However a larger study looking at people with isolated ACL is needed.

•The study participants were recruited from Imperial College London and Imperial College Healthcare NHS. We recruited via posters around the campus, unions and gyms. We were able to match for gender and activity (using tegner activity scale) but not age. However this is no longer found to be statistically significant.

Abstract (Britt Elin Oiestad)

-Background: the aim of the study is unclear stated in the background. Please specify according to investigating changes in gait, and not knee function.

•Agreed, we have clarified our primary and secondary aims here. Page 2 line 14 – 21.

-Methods: It would be informative if the gait measures were specified. Type of statistical methods?

•Further clarification of gait parameters are now included and we have specified that it was a 'MANOVA that was performed to determine statistical significance.' Page 2 line 39 – 42

-Results: In the sentence "A similar, but non-significant trend was seen in all the other activities": please specify the activities.

•'A similar, but non-significant trend was seen in slow, uphill and downhill gait.' Page 3 line 3 – 6

-Conclusion: the first part of the conclusion is based on 5 subjects with isolated ACL injury.

Furthermore, only 7 subjects had additional injury. On the basis of the results presented in the abstract, the conclusion seems too bold.

•This has been modified to 'may lead to a higher predisposition.' Page 3 line 24

-In the article summary: new information in bullet 2 that is not mentioned in the abstract is given. This should be implemented in the method section and the results.

•We have implemented them into methods accordingly. Page 2 line 37 – 39

Introduction (Britt Elin Oiestad)

-Page 5, line 3: is the avoidance pattern only to prevent further ACL damage, or is it also because of quadriceps activation failure?

•It is thought that quadriceps activation failure is due to a central regulatory attempt by the knee to avoid further damage to the joint, by inhibiting these muscle groups. We have made this clearer in the manuscript. Page 5 line 8 – 10

-Page 5 line 20: do the authors mean 1% increase in adduction moment (not "adduction of moment")?

•We have clarified this to adduction moment.

-Page 5, line 25-30: the aims stated here is different from the aims in the abstract. Please be consistent with the aims throughout the manuscript.

•This has been modified. Page 5 line 37 – 51

Methods (Britt Elin Oiestad)

-How was the study participants included? And from where? Which time point after ACLR were the study participants examined?

•Please see above. The mean time since their ACLR was 4.5 years.

-Page 5, lines 47-57. Please re-write to include more sentences and fill in missing words. I prefer subject characteristics in the beginning of the result section, and a presentation of methods only, in the method section.

•We have removed subject characteristics from the methods section and adjusted accordingly.

-Page 6, line 15: the control participants seem somewhat older. Why weren't the control subjects' matched on sex, age and activity level?

•This was done as a BSc project, due to time pressures and recruitment through posters it was difficult for us to directly match our groups.

-Page 6, line 23-37: Adding up the markers described does not give a total of 35 as indicated in line 28. Please be specific if the number 35 means the capacity of the camera or the amount of markers that were included in the study.

•The clusters had 3 markers on each, we have made this clearer in the manuscript. Page 6 line 57

-Page 6, line 38: why were reflective markers put on acromion and manubrium?

•These markers were for us to have a sense of the subjects upper body positioning and were not used in the model.

-Page 7, line 40-50: please include power calculations for rationale of the amount of subjects included in the study.

•Roy's Largest Root found the power of the multivariate tests looking at gait outcome to be 0.838.

-Page 7, line 41: Why do you use paired sample t-test when there is no matching of the cases and controls? Are the sample correlated in any way?

•We should not have used paired t-test, thank you for pointing this out. We have changed this to unpaired t-test.

Results (Britt Elin Oiestad)

-Page 8, line 6: why were the controls older than the cases? Or put another way, why weren't the controls matched on age? Please give an explanation.

•Please see above.

-Page 8, line 6-11: Which variable were included in the multiple regression analyses to assess the influence of age? Did both the controls and cases be accounted for in this analysis?

•Variables included age, weight, height and activity. Yes the controls were accounted for in this analysis. However we have now removed this, as difference in age was not found to be statistically significant.

-Page 8, line 13: the Tegner Activity Scale should be included in the method section as an outcome measure.

•We have included this in the methods. Page 6 line 35 – 37

-Page 8, line 36-44: The groups are very small. Give a rationale for why the sample is divided into groups.

•This was an exploratory study. Only once we were analysing the results did we notice the discrepancy between those with isolated ACL tear and those with ACL tear with additional injuries.

-Page 8, line 34: I suggest answering the aim of the study starting from here, and present less important results that are not stated as aims of the study in the end of the result section.

•We have placed the less important results at the end of this section.

-Page 9, Table 2: it is not clear what the column of «Post hoc analyses» means. Please explain under the table.

•Post hoc analysis is showing which of the results were found to be significant. We have changed this to significant differences.

-Page 9, Table 2: It is unclear what the p-value reflect (for which groups?). If the p-values reflect ACLR vs. controls, the table would be more visible if the p-values were moved to fourth column. The same goes for table 3. Please also consider to make a table for the gait variables as table 2 – the most important results should be presented before the functional data which are not specified in the objective of the study.

•We have removed the P-value column and added p-values in the significant values column. We have now presented gait outcomes before the functional data, however we have displayed the gait outcomes using our figures.

Discussion (Britt Elin Oiestad)

-I would suggest that the first paragraph of the discussion present the aim of the study, and the results of the study.

•We have changed our discussion accordingly. Page 10 line 40 – Page 11 line 27.

-Page 10, line 37: the reference Claes et al. should be updated to Oiestad et al 2009, which is the original review of the numbers presented here.

•We have updated this reference. Oiestad BE, Engebretsen L, Storheim K, Risberg MA. Knee osteoarthritis after anterior cruciate ligament injury: a systematic review. Am J Sports Med 2009;37:1434–1443 – REF 24.

-The statement on page 10, line 42-47 «Our results suggest that high knee adduction moments may be a relevant risk factor in OA incidence in ACLR subjects only when associated knee injuries are present.», should be modified or deleted. This study have no data on osteoarthritis, thus, no association between knee adduction moments and OA can be made.

•We have removed this statement.

-Line 55: where did the authors present data to underline the statement of a difference between ACLR+ and controls?

•This tendency can be seen in figure 2.

-Page 10, line 55 and page 11, line 2: please replace “to controls” with “compared to controls”

•Thank you, we have changed this

-Page 12, Line 38-39: use KOOS consistently

•We have corrected this.

-The written English should be updated on some parts of the manuscript.

•We have attempted to revise the grammar throughout.

-The STROBE statement is attached, but it is not mentioned in the manuscript.

•We have included this in our methods.

Reviewer 2 (Zi-Sheng Ai)

-Please explain why not consider age, gender as matching factors?

•Please see above.

-In Page8 Line 25, after age differences appeared in the two groups of Table1, Please explain the rationality of using One-way ANOVA in Table2 & Table3?

•Thank you for pointing out this mistake, we have now changed the statistical test to MANOVA. This did not change any statistical differences.

-Statistical analysis are carried out using the SigmaPlot 11.0 on windows, please explain why not used commonly SAS, SPSS or Stata software?

•SigmaPlot 11.0 was the available statistical software at the time, however we have now redone all the statistical tests on SPSS.

-In this study, the sample size is too low, the conclusion is not reliable, please calculate power of a test?

•Roy's Largest Root found the power of the multivariate tests looking at gait outcome to be 0.838.

-Table2 in Page9 Line 3, and Table3 Page9 Line40, there were one-way analysis of variance among four sample means, which ACLR, itself including ACLR + and ACLR-. It was wrong about carrying out one-way ANOVA of comparing four sample means & multiple comparison in statistics, please correctly analyze the data.

•We have carried out MANOVA statistical test instead on SPSS. This did not affect our results.

-Furthermore, patients with ACL injury were divided into two subgroups. In ACLR+ group, we find there are 3 cases with associated meniscal tear, 3 with cartilage damage and 1 with torn MCL. That means, the ACLR+ group is heterogeneous. We do not know which associated injury is more important. Therefore, I think the conclusion is not supported totally by the methodology.

•Conclusion has been modified to reflect this.

-Language: Good. However, needs a more careful proofreading. There are some grammar mistakes and in some points the content is confusing. Page 13 line 46 "artefacts " need a revision.

VERSION 2 – REVIEW

REVIEWER	Zi Sheng Ai College of Medicine, Tongji University, Shanghai, China
REVIEW RETURNED	14-May-2014

- The reviewer completed the checklist but made no further comments.